# Position: Evaluations Should Acknowledge Model Multifacetedness in the Era of Large Language Models

## Abstract

The rapid evolution of Artificial Intelligence (AI), particularly Large Language Models (LLMs), marks a significant departure from earlier machine learning (ML) paradigms. This advancement has exposed critical misconceptions in our understanding of the "model" itself, especially evident in evaluation methodologies that often rely on narrow observational windows to assess overall model quality. This paper argues that a fundamental reconceptualization of the "model" itself is necessary to address this evaluative crisis. We introduce a five-tiered hierarchical framework. Specifically, we divide models into: *Noumenal*, *Conceptual*, *Instantiated*, *Reachable*, and *Observable* ones. Using this framework, we examine the historical development of how models have been conceptualized and evaluated within the ML field, analyzing the roles of experiments, ablation studies, and datasets. The paper further argues that LLMs' current development fundamentally challenges these long-standing evaluation patterns, as existing benchmarks and metrics increasingly fail to capture the true capabilities and limitations of these complex models. Our primary contribution is to consolidate and structure many of these historical insights and evolving challenges. By organizing these often fragmented pieces of understanding into the proposed five-tiered hierarchical framework, we aim to offer a more cohesive and systematic lens for approaching AI model evaluation. We believe that such a structured approach, which encourages assessment strategies to be explicitly contextualized by a model's position within this hierarchy and informed by its preceding layer, can help cultivate a more robust and meaningful comprehension of these increasingly complex LLM systems.

## 1 Introduction

Artificial intelligence has undergone several phases of rapid advancement. Yet, the recent emergence and widespread adoption of Large Language Models mark a fundamental shift [11, 17, 33, 36, 132]. This development significantly challenges established approaches, not only in how AI is created but, crucially, in how it is evaluated [5, 21, 67, 71, 110, 127]. The sophisticated capabilities of contemporary AI models have surpassed the existing conceptual and methodological tools previously used to understand and evaluate ML systems [11]. This paper will explore the characteristics of this significant change, by proposing a hierarchical perspective for assessing AI models, to help navigate the present difficulties in LLMs' evaluation. Historically, earlier ML systems were typically designed for clearly defined, narrow tasks, such as classifying images or detecting spam [61]. While their internal structures could be complex, they were often more transparent. Evaluation metrics could frequently provide a direct measure of the model's usefulness for its intended function. For example, the accuracy of a classification model was a relatively clear indicator of its performance [10]. In contrast, LLMs are moving beyond restricted, task-specific roles towards more general abilities and often display emergent behaviors [121]. These new emergent skills—such as learning from examples

within the prompt (in-context learning [16]), step-by-step reasoning (chain-of-thought [122]), and even behaviors that appear to be creative or strategic [45, 128], were not the primary goals of their design, nor were they easily observable in older ML models. Often, the full range of their potential behaviors is not known in advance, even by their creators [11]. The appearance of unexpected abilities signifies that a system's overall properties cannot be fully understood simply by examining its individual components or its initial design specifications [3].

Currently, the most common approach to assessing LLMs is through what might be called a "small observation window". This method usually involves testing models on standardized benchmarks. However, by their very nature, these benchmarks can only examine a tiny fraction of a model's potential range of behaviors [96]. Such limited observation can lead to an erroneous understanding regarding a model's true quality, its ability to generalize to new situations, and its potential risks, including safety concerns [49]. For instance, high performance on a specific benchmark might result from issues like data contamination [72] (where test data was accidentally part of the training data) or it might simply show that the model has overfitted to the particular behavior of the benchmark tasks, rather than demonstrating a genuinely robust and widely applicable capability. **Therefore, evaluating LLMs as if they were merely more powerful versions of traditional, fully understandable ML models is a fundamental error in *categorization*** [101]. An over-reliance on these narrow evaluation windows can inadvertently create a superficial or misleading impression of understanding, akin to a "simulacrum" [6]. In this situation, reported benchmark scores can become disconnected from the model's actual abilities. They may represent a performance specifically *manufactured* for that benchmark, rather than an intrinsic, generalizable quality of the model itself. This practice can result in a hyperreal [125] assessment environment within the research community, where the benchmark score is treated as more significant or *real*. Such a scenario risks skewing the research agenda towards optimizing performance on these limited benchmarks, rather than pursuing a more comprehensive understanding or development of AI capabilities.

While much of the existing literature has concentrated on the design and refinement of evaluation benchmarks [13, 21, 22, 25, 26, 52, 59, 64, 66, 76, 86, 93, 96, 98, 99, 108, 110, 116, 130, 133], this paper seeks to complement these efforts by focusing on the underlying conceptualization of the *model* itself. We observe that effectively addressing the current challenges in evaluating LLMs **benefits from a clearer and more structured understanding of what constitutes a "model" in this evolving landscape**. Our work aims to synthesize various perspectives by proposing that models can be understood across multiple levels of abstraction and concrete realization. To this end, we introduce a five-tier hierarchical framework, which forms the conceptual backbone of this paper:

- **Noumenal Model**: The ultimate, and perhaps inherently unknowable, generative principles or reality that the AI system is intended to approximate or capture.

- **Conceptual Model**: The intended design, underlying theories, and architectural blueprints.

- **Instantiated Model**: The actual implemented algorithmic artifact with an initialization state.

- **Reachable Model**: The optimized model, with the full spectrum of its potential behaviors.

- **Observable Model**: A subset of behaviors that are actually witnessed during specific evaluation procedures and interactions.

The subsequent sections will discuss the definition of these five tiers, drawing inspiration from established traditions of modeling and abstraction in both philosophy and science [37]. We will then trace the historical development and relevance of these conceptual layers within the field of machine learning, highlighting how the emergence of LLMs overturns the interrelation between these layers. Finally, we propose that robust assessment should involve a more deliberate and structured approach, where evaluations conducted at a specific model tier are explicitly defined and constrained by an understanding of the preceding, more fundamental tier, contributing to a more systematic framework for meaningful, comprehensive, and reliable evaluations of LLM systems.

## 2   Hierarchical Ontology of Models

To navigate the complexities of modern AI systems, particularly LLMs, it is proposed that the very concept of "model" be deconstructed and reassembled into a hierarchical ontology. Before the detailed definitions, we first briefly introduce our inspiration drawn from established principles in system theory, cognitive science, AI planning, and machine learning.

Hierarchical analysis of complex systems is a well-established paradigm. General system theory [12, 114] and hierarchy theory [106] explain that layered structures improve understanding of system components, interactions, scales, and observer roles, helping manage complexity due to differing interactions and emergent properties across levels. Such analysis is also applied in cognitive science for AI frameworks. For example, David Marr proposes three levels for understanding information-processing systems, the computational theory (goal and logic of computation), representation and algorithm (how the theory is implemented), and hardware implementation (physical realization) [75]. Similarly, Allen Newell distinguished between the knowledge level, which describes a system in terms of its goals and the knowledge it rationally employs, and the symbol level, which details the specific symbolic representations and processes that mechanize this knowledge [84, 85]. These frameworks highlight the importance of differentiating between abstract purpose, procedural specification, and concrete instantiation. Furthermore, fundamental concepts from ML also inform our hierarchical view. For instance, the initial design of an AI system often implicitly defines a hypothesis space for all the possible functions or solutions [10, 81], thus an algorithm could then search this space within a specific instantiation. Crucially, these ML theories also emphasize the distinction between a model's true generalization capability (on unseen data) and its observed performance on finite test sets. Building upon these diverse theoretical foundations, our proposed five-tier hierarchical ontology aims to provide a specialized framework tailored to the nuances of modern AI systems, particularly LLMs, and the challenges they pose for evaluation.

**Definition 2.1 ($\mathcal{M}_N$: *Noumenal*)** *The Noumenal Model represents the ultimate, perhaps intrinsically unknowable, generative principles or the "true" underlying structure of the reality that an AI system aims to capture or approximate. The Noumenal Model is the ideal form of knowledge or the perfect causal understanding of a domain.*

Philosophically, this concept draws inspiration from Immanuel Kant's notion of the *noumenon* or *thing-in-itself* [57], particularly his distinction between *phenomena* and *noumena* (*Critique of Pure Reason*, A235/B294–A260/B315). We can conceptualize a theoretical machine learning model that remains fundamentally unrecognizable to human beings, and which we can only ever imperfectly apprehend through phenomena [73]. Such a model would not be a *black box* whose mechanisms are too complex for us to trace, but rather one whose fundamental operational principles and cognitive architecture have no common standard of human thoughts and empirical observation.

On one hand, the existence of $\mathcal{M}_N$ is in the fundamental assumptions in the philosophy of science, which posit an objective reality governed by (perhaps not fully) discoverable and comprehensible natural laws (through systematic observation and experimentation). On the other hand, though *wholly unknowable*, recognition of a $\mathcal{M}_N$ carries practical weight, compelling critical examination of AI's fundamental goals. For instance, contemporary LLMs are primarily trained to predict the next token in a sequence, implicitly adopting the data's statistical distribution as their learning target. However, if $\mathcal{M}_N$ truly incorporates profound principles such as "core knowledge" [60, 109] or "causal structures" [90], then merely mimicking surface-level statistical patterns in data may be insufficient, resulting in the *brittleness* of LLMs. Consequently, holding the idea that any scientific system can only provide an approximation of the $\mathcal{M}_N$, encourages a re-evaluation of AI's ultimate objectives and the methodologies used for designing the learning tasks.

**Definition 2.2 ($\mathcal{M}_C$: *Conceptual*)** *The Conceptual Model comprises the intended design and specified architecture, underlying theory and theoretical assumptions, chosen algorithms and blueprint of the system, and finally, the high-level goals the system is meant to achieve, as envisioned by its human creators.*

Following the Kantian inspiration, the human mind actively structures experience through a priori categories of understanding (*e.g.*, causality, unity) to make sense of the phenomenal world (*Critique of Pure Reason*, B1-B2, A70/B95-A83/B109). $\mathcal{M}_C$, therefore, imposes a conceptual structure onto a problem domain or desired functionality, from the observed phenomena. More specifically, it contains i) the system's high-level objectives (*e.g.*, the form of loss functions), ii) the theoretical assumptions guiding its operation (*e.g.*, assumptions about the data, learning processes), iii) the selected algorithms and data structures, iv) the overall formal description of the system which act as Kantian schemata that mediate between pure concepts and observations.

$\mathcal{M}_C$ is a necessary abstraction (*e.g.*, "attention"), with logic formalizing it in AI systems (*e.g.*, "is all you need" [112]). The logical framework enables structured human thought to engage with complex

realities, allowing designers to specify an AI's intended knowledge, reasoning, and behaviors [14, 100]. Although the logical formalisms of the abstracted $\mathcal{M}_C$ may not fully predict or constrain the complex behaviors of these systems in operation (especially, LLMs' actual behaviors can largely diverge from an (expectative) logical rigor design, see Section 3). Nevertheless, acknowledging its limitations does not diminish the importance of $\mathcal{M}_C$; it constitutes the logical starting point, becoming the vitally important reference benchmark for evaluating behavior deviation, diagnosing system failures, and understanding unexpected problems.

**Definition 2.3 ($\mathcal{M}_I$: *Instantiated*)** *The Instantiated Model refers to the actual, concrete algorithmic artifact that has been implemented in code and exists as a computational entity, encompassing the specific implementation of algorithms, the precise network architecture, the initialized parameter values, and the exact software and hardware environment in which the model operates.*

We *intentionally* define the concept of *initialization parameters* more vaguely and expansively, encompassing a potential pre-training phase (at any specific checkpoint, but before task-specific fine-tuning), not just a single random initialization of an established network. This is because the initialization scheme itself also constitutes a concrete instantiation of the $\mathcal{M}_C$'s abstracted content. For instance, a neural network could be initialized (and further optimized) randomly [102], orthogonally [54], or self-supervisedly with a large-scale dataset [16, 31]. These initialized parameter values define the model's specific state at a particular stage, directly influencing its subsequent learning trajectory and potential capabilities (of the Reachable Model). For instance, a pre-trained $\mathcal{M}_I$ can be highly structured, with parameters encoding significant general-purpose knowledge and representations. Indeed, parameters taken from any specific checkpoint during or after a training process also define a distinct $\mathcal{M}_I$, a snapshot of its learned state. However, it is crucial to distinguish $\mathcal{M}_I$ from merely *a pre-trained model*, despite being a key example due to their structured initial parameters. $\mathcal{M}_I$ more broadly signifies the model's tangible, concrete configuration at any defined starting point that serves as the foundation prior to the specific optimization process designed to evolve it towards its Reachable counterpart.

Furthermore, the specific characteristics of $\mathcal{M}_I$ play a crucial role in constraining and shaping the subsequent Reachable Model. The journey from the $\mathcal{M}_C$ (*e.g.*, the idea of attention mechanism) to the $\mathcal{M}_I$ (*e.g.*, the specific code with initial weights of a Transformer) involves numerous design choices and initial conditions. Small variations in architecture or minor differences in initialization can send the model down different optimization paths, leading to distinct Reachable Models ($\mathcal{M}_R$) with varying capabilities and biases. This is a critical juncture, as these early decisions and their non-obvious influences on the model's development represent the first steps in *a gradual departure from the original concept*, significantly contributing to the well-known "black-box" problem [68]. Nevertheless, gaining a better understanding of the $\mathcal{M}_I$'s intrinsic properties (its architecture, representational style, and initial state) is critical for anticipating the characteristics of the final, trained Reachable Model.

**Definition 2.4 ($\mathcal{M}_R$: *Reachable*)** *The Reachable Model is the Instantiated Model after its optimization on a specific learning dataset (i.e., the set of finalized learned parameters). More broadly, it encompasses the full spectrum of potential behaviors and internal stochastic processes (e.g., sampling strategies) that the optimized model could exhibit across all possible valid inputs.*

In general, $\mathcal{M}_R$ signifies more than just a *post-trained model*. While the "Reachable" materializes after an optimization process acting upon an $\mathcal{M}_I$, its defining characteristic is the representation of the model's complete potential capabilities, a direct consequence of its specific learned parameters. Thus, the focus is on this entire accessible behavioral repertoire, rather than merely the model's status as having completed a training phase. While $\mathcal{M}_R$ represents the totality of what the model can ultimately do, much of this capacity may not be immediately apparent from its static components or the original design intentions. Meanwhile, the inscrutable nature of the training process further intensifies the departure of $\mathcal{M}_R$ from the initial concept. Consequently, $\mathcal{M}_R$ becomes more akin to what is typically understood as a "black-box model." Furthermore, it is within $\mathcal{M}_R$ that *emergent abilities* manifest, which were not explicitly designed into $\mathcal{M}_C$ nor readily predictable from the $\mathcal{M}_I$ alone, but arise from the interplay of scale, data, and the optimization process. Such behavior is indeed central to the essence that the term "black box" seeks to embody, while significant prior research in this domain has already been dedicated to understanding $\mathcal{M}_R$. Examples include work on adversarial testing [43], red-teaming [39], and frameworks for predicting emergent abilities [121]. This underscores that critical aspects like AI safety [49] and alignment [4] are, at their core, attributes

201  of $\mathcal{M}_R$, necessitating evaluation strategies far more comprehensive than current standard practices
202  and equipped to grapple with its inherent complexity and opacity.

203  **Definition 2.5 ($\mathcal{M}_O$: *Observable*)** *The Observable Model constitutes the subset of the Reachable*
204  *Model's behaviors that are* actually *witnessed, measured, and documented through available/existing*
205  *evaluation protocols, datasets, and metrics. The Observable Model is the empirical manifestation of*
206  *the AI systems' performance under particular inspection.*

207  The observable manifestation is precisely what current AI benchmarks aim to capture, for instance, in
208  the natural language process scenarios, we use MMLU [48, 120] for general knowledge, GLUE [116]
209  and SuperGLUE [115] for natural language understanding, and more comprehensive frameworks
210  like HELM [66]. However, the critical issue is that the choice of what to observe profoundly shapes
211  our perception of an AI's capabilities. This is because how convincing (*plausibility* [32, 79]) an
212  explanation of an observed behavior is to a human user is often based on interactions with, and
213  interpretations of, the Observable Model. For example, if an LLM is observed to perform well
214  on simple problems presented in a benchmark but fails on more complex versions of the same
215  underlying task, then we probably recognize this LLM as having only primary capabilities on this
216  task, which can be a total misunderstanding about the potentiality resided within the Reachable
217  Model. Unfortunately, essentially, even though current benchmarks have been working hard on
218  providing a better observation window. For instance, HELM strives for "Broad coverage... Multi-
219  metric measurement... Standardization" to improve how the Observable Model is captured. They
220  still need to *explicitly* acknowledge the inherent incompleteness of any such observation. In this way,
221  the Observable Model can become a skewed or unrepresentative sample of the Reachable Model's
222  true nature, and optimizing for it does not necessarily translate to the underlying Reachable Model
223  having improved in a broadly generalizable manner, nor does it guarantee closer approximation to
224  the Conceptual or Noumenal ideals.

## 3  Evolution of Model Conceptualizations

226  The conceptualization of the "model" in machine learning has not been static; rather, it has undergone
227  a continuous process of evolution and enrichment. The hierarchical structure situated above the
228  Noumenal Model, was not an instantaneous creation, nor did it arise *spontaneously* with current
229  advanced systems like LLMs; rather, it reflects a gradual process of differentiation. Specifically, when
230  an AI system has significantly expanded the scope of its capabilities and conceptual complexity, a
231  more concrete model tier would be "crystallized" from the lower one. Below, we will demonstrate this
232  change through a rough definition of $>$ and $\simeq$ between models of different tiers. Briefly, Model Tier
233  A $>$ Model Tier B (A is broader/encompasses B) signifies A is more fundamental, B is a constrained
234  version or subset of A, and the A-to-B transition involves reduction or constraint. Model Tier A $\simeq$
235  Model Tier B (A is similar/equal to B) signifies no significant *practical gap* between them; they
236  largely capture each other reciprocally, and transitioning between them doesn't substantially alter
237  information or their core nature.

238  **Differentiation from the Conceptual Model**: For models such as Naive Bayes and Decision Trees,
239  which possess relatively simple structures and clear theoretical underpinnings, their hierarchical
240  relationship can be expressed as: $\mathcal{M}_N > \mathcal{M}_C \simeq \mathcal{M}_I \simeq \mathcal{M}_R \simeq \mathcal{M}_O$. For instance, if the ultimate
241  true principles of the target domain, *e.g.*, the true biological mechanisms for disease prediction,
242  are represented in $\mathcal{M}_N$. But, the Naive Bayes classifier based on selected features for disease
243  prediction [46] represents a simplified concept, capturing only a limited, abstracted view of the
244  observation, often with strong independence assumptions [10]. Meanwhile, these simpler models'
245  $\mathcal{M}_C$ can be generally translated into $\mathcal{M}_I$'s implementation faithfully, since there are fewer degrees
246  of freedom that would lead to significant deviations. For instance, the recursive partitioning logic
247  and splitting criteria for decision tree algorithms like ID3 or C4.5 [81, 95] strictly follow the concept
248  of a tree structure. The training process then fully determines $\mathcal{M}_I$'s final form and behavior, such
249  as calculating conditional probabilities for Naive Bayes from data, or selecting splits and growing
250  branches for a decision tree. Since these models operate based on explicit, inspectable rules or clearly
251  defined probabilistic inferences [82], the space of potential outputs for any given input is constrained
252  and directly calculable from the $\mathcal{M}_R$. Finally, due to this deterministic and transparent nature, $\mathcal{M}_R$'s
253  full spectrum of potential behaviors can be comprehensively captured by standard evaluation metrics
254  (*e.g.*, precision, recall, F1-score, ROC curves) on representative test sets. Therefore, we conclude that
255  $\mathcal{M}_O$ derived from such evaluations is thus a reliable and sufficiently complete representation of $\mathcal{M}_I$
256  and $\mathcal{M}_R$'s capabilities and limitations for the defined problem scope.

**Differentiation from the Instantiated Model**: For models like K-Nearest Neighbors (KNN), Support Vector Machines (SVM), and Linear Regression, their hierarchical relationship shows a subtle change: $\mathcal{M}_N > \mathcal{M}_C > \mathcal{M}_I \simeq \mathcal{M}_R \simeq \mathcal{M}_O$. While $\mathcal{M}_N$ of ultimate reality transcends any human-designed $\mathcal{M}_C$, a key distinction is that the theoretical ideals within the $\mathcal{M}_C$ are more abstract than their practical implementation. For instance, SVM's maximum-margin hyperplane and the kernel trick [10, 28], Linear Regression's best-fitting plane achieved by minimizing a loss function [46], or KNN's neighbor-based decision principle [29, 46]. This separation occurs because instantiation necessitates specific, constraining choices that are not contained in underlying concepts, such as particular SVM kernel functions (*e.g.*, RBF, polynomial) and regularization parameter [104], optimization algorithms and loss functions like SMO [92] for SVMs or gradient descent with L2 regularized mean squared error for linear regression, or defined K-values and distance metrics (*e.g.*, Euclidean, Minkowski) for KNN. These choices make the implemented algorithmic artifact ($\mathcal{M}_I$) a particular, constrained realization of the broader conceptual theory. Despite this $\mathcal{M}_C > \mathcal{M}_I$ distinction, once these models are trained and their parameters are finalized (*e.g.*, support vectors identified, regression coefficients determined, or training samples stored for KNN), their behavior becomes fully determined by this learned state, since there are generally no further complex emergent abilities beyond what is directly implied by the chosen structure and learned parameters. Furthermore, these instantiations, even with specific choices, are still highly structured and predictable. Their mechanisms are transparent enough (*e.g.*, linear coefficients, support vector locations, distance calculations) to allow standard evaluation methods to comprehensively capture their performance on test data, making $\mathcal{M}_O$ a faithful and reasonably complete representation of $\mathcal{M}_R$.

**Differentiation from the Reachable Model**: For models like Shallow Neural Networks (Shallow NN), Multilayer Perceptrons (MLP), and Restricted Boltzmann Machines (RBM), the distinctions between tiers intensify further, typically expressed as: $\mathcal{M}_N > \mathcal{M}_C > \mathcal{M}_I > \mathcal{M}_R \simeq \mathcal{M}_O$. The gap widens from $\mathcal{M}_I$ to $\mathcal{M}_R$, as this tier critically encompasses not only the specific architectural implementation (*e.g.*, topology of a three-layer MLP and choice of activation functions) but also the initial parameter values (*e.g.*, random initializations [41, 47]), which are vital for the subsequent optimization trajectory as they set the starting point in a complex, non-convex loss landscape [69, 77]. Consequently, the complex optimization process of training a neural network transforms the initial states ($\mathcal{M}_I$) to ones with significantly different capabilities and behaviors ($\mathcal{M}_R$). Different initialization seeds [91] or minor variations in the optimization process [20] can lead the network to converge to different local minima in the loss landscape, resulting in distinct $\mathcal{M}_R$ even from nearly $\mathcal{M}_I$ Models [42]. Nevertheless, for these shallow networks, although their internal representations may begin to exhibit the opacity characteristic of deep learning (*i.e.*, having global non-local sub-representations) [87], their overall behavioral complexity is generally considered sufficiently bounded. Practically, it is often assumed that standard, diverse benchmarks and evaluation metrics can still capture their core capabilities and generalization performance reasonably well [46], making $\mathcal{M}_O$ a fair, albeit perhaps not exhaustive, representation of $\mathcal{M}_R$'s overall performance.

**Differentiation from the Observable Model**: As network depth and complexity increase, Deep Neural Networks (DNNs) exhibit more intricate hierarchical relationships, summarized as: $\mathcal{M}_N > \mathcal{M}_C > \mathcal{M}_I > \mathcal{M}_R > \mathcal{M}_O$. While the distinctions established in shallow NNs persist, a critical new divergence distinguishing DNNs arises between $\mathcal{M}_R$ and $\mathcal{M}_O$. Due to their vast parameter counts, deep architectures, and extensive training on large datasets, DNNs learn extremely complex functions, resulting in an $\mathcal{M}_R$ with an enormous potential behavioral space not explicitly programmed nor easily predictable from $\mathcal{M}_I$ alone. However, our current methods of observation, standard evaluation protocols and benchmarks such as ImageNet [30], GLUE [116], or even comprehensive frameworks like HELM [66], can only access a limited subset of this vast behavioral repertoire. $\mathcal{M}_O$ is frequently reported to fail in fully presenting the true scope of $\mathcal{M}_R$'s capabilities. For instance, models' brittleness is easily demonstrated when faced with out-of-distribution inputs or slight adversarial paraphrases, which exposes superficial "shortcut" learning rather than robust understanding [40, 49, 86, 96].

**A note on large language models**: For contemporary LLMs, the hierarchical gaps between conceptual tiers of models are widening dramatically, with the largest and most significant divide occurring between $\mathcal{M}_R$ and $\mathcal{M}_O$, broadly expressed as: $\mathcal{M}_N > \mathcal{M}_C > \mathcal{M}_I > \mathcal{M}_R >> \mathcal{M}_O$. While significant gaps separate Noumenal model goals (*e.g.*, representing human language, knowledge, and reasoning) from Conceptual designs and Instantiations (*e.g.*, Transformers [112], Mamba [44]). The pre-existing distinctions are amplified in $\mathcal{M}_R$ created by extensive post-training. $\mathcal{M}_R$ of an

LLM exhibits an immensely vast potential behavioral space, featuring (sometimes) unpredictable emergent abilities like in-context learning, instruction following, and complex reasoning [119, 121]. Concurrently, significant risks are reported, such as generating hallucinations [55], amplifying biases [7], or producing harmful content [123]. However, the combinatorial nature of language and the sheer scale of these models create a serious mismatch; what we learn from $\mathcal{M}_O$ is a very incomplete picture of an LLM's true overall abilities and the hidden dangers within its $\mathcal{M}_R$. This mismatch is a fundamental reason for the current problems in testing LLMs, the major challenges in making them behave safely and as intended, and the troublesome practice of "SOTA chasing".

This evaluation challenge appears to be significantly compounded by the field's tendency to rely on an evaluation paradigm inherited from earlier ML. In those earlier and simpler systems, relative transparency and tighter coupling between the tiers characterized the confidence in standard metrics and experimental setups. These approaches became deeply ingrained and are now being somewhat uncritically applied to LLMs. With LLMs, the relationships between the tiers have become significantly more complex, opaque, and divergent. The historical success of these evaluation norms with simpler models established certain "patternized experiments" and expectations about what constitutes "good evaluation." These established practices were then naturally carried over when LLMs emerged, despite them possessing vastly different characteristics, particularly in the complexity and opacity of their Instantiated and Reachable tiers. **This "historical muscle memory" from evaluating simpler models, when applied to the new context of LLMs, can be seen as a significant contributor to the current evaluation challenges.** In many ways, the field might have been attempting to navigate new, complex terrains using guides developed for older, more familiar landscapes.

## 4 Towards Hierarchically-Informed Evaluation Strategies

The advent of LLMs necessitates a significant re-evaluation of established methodologies for experimental design, ablation studies, and dataset curation, which form the bedrock of traditional "patternized experiments." This situation demands a shift in the overarching goal of LLM evaluation: moving away from rendering final, summative judgments based predominantly on leaderboard rankings ($\mathcal{M}_O$) towards an ongoing, iterative process of *model cartography*. To achieve this, we first review how LLMs break the patterns, then we introduce the paradigm that hierarchically probes different aspects across model tiers, by assessing each in explicit relation to its antecedents. A detailed proposal for this frameworks is provided in the appdenix.

### 4.1 Rethinking Experimental Design, Ablation, and Dataset Curation

**Experimental design** aimed to test hypotheses derived from an $\mathcal{M}_C$ (*e.g.*, the utility of a specific feature) or to generate an $\mathcal{M}_O$ by measuring a trained $\mathcal{M}_I$'s performance. While typically, experimental designs for early ML assume variables could be isolated and their effects clearly measured, this paradigm is inadequate for LLMs. Their profound sensitivity to subtle variations in prompting [70] and the immense difficulty in controlling for confounding variables when assessing complex, generative behaviors make controlled experiments challenging. Static benchmarks, forming the traditional $\mathcal{M}_O$, often fail to capture the dynamic and vast capabilities of an LLM's $\mathcal{M}_R$. Moreover, such benchmarks cannot explicitly probe inter-tier relationships (*e.g.*, assessing the fidelity between a Conceptual design choice, like an architectural modification for long-context reasoning, and its actual manifestation in $\mathcal{M}_I$'s representations), or systematically explore hypothesized regions of the Reachable behavior space to understand emergent capabilities [121] and their operational boundaries.

**Ablation studies** have historically served to understand component contributions within an $\mathcal{M}_I$ to its Reachable performance, helping to validate or refine $\mathcal{M}_C$ choices [42]. This reductionist approach, however, faces severe limitations with LLMs. In these highly complex, non-linear systems, representations are often distributed, and components (like neurons or attention heads) can be polysemantic, contributing to multiple functions [34, 88]. Consequently, removing (or changing) a component from an LLM doesn't simply isolate its original function; it can yield a fundamentally different system with altered internal dynamics and potentially different emergent properties. Interpreting such changes as the "contribution" of the ablated part becomes problematic, akin to challenges in intervening on complex causal systems [90]. Simple ablations may therefore offer superficial or even misleading insights into an LLM's $\mathcal{M}_I$'s functional architecture or the mechanisms generating its Reachable behaviors. Therefore, more cautious interpretations are essential, and complementary approaches like perturbation studies (assessing sensitivity to small changes) or influence studies (tracing impacts

of training data/features) might offer more reliable, albeit still partial, insights into these deeply interconnected systems.

**Dataset curation** focused on gathering data samples presumed to be representative of some aspect of the *Noumenal Model*. The goal was to train an $\mathcal{M}_I$ into a generalizable $\mathcal{M}_R$, carefully navigating the bias-variance tradeoff [10]. However, in the LLM scenarios, the scale and often uncurated nature of web-derived training corpora make it exceptionally difficult to ensure true representativeness or, critically, to prevent contamination with data that might overlap with evaluation benchmarks. Such contamination can grossly inflate $\mathcal{M}_O$ performance, providing a misleading picture of an LLM's real generalization capabilities within its Reachable space [72]. Therefore, datasets should essentially be tools for characterizing $\mathcal{M}_R$ itself, verifying alignment between Conceptual and Instantiated tiers.

## 4.2 Paradigm of Model Cartography

**Beyond Current Benchmarks for the Observable Model**: $\mathcal{M}_O$'s performance and behaviors should be interpreted not in isolation, but in direct relation to the known (or reasonably estimated) properties of $\mathcal{M}_R$. The key question becomes: Does the observed performance on a benchmark or specific task reflect a robust and generalizable capability within the broader Reachable space, or is it an isolated success, perhaps an artifact of the evaluation setup, data contamination, or a highly specific and narrow competence? This involves probing the gap between observed performance and latent potential. Therefore, the primary goal is to determine if observed behaviors are indicative of broader, stable capabilities within the Reachable space or are mere artifacts. To achieve this, we should consider the following:

- Utilize metrics like the Model Utilization Index [19] to assess if high Observable performance stems from robust, general mechanisms (indicating broad Reachable capability) or overused, narrow circuits (implying fragile, benchmark-specific success).

- Employ adaptive testing, dynamic benchmarks, and interactive protocols to explore the Reachable space, especially around areas of success or failure identified in static tests [18].

- Focus on "construct validity" [2]: investigate if Observable benchmark performance correlates with diverse, real-world task performance when both theoretically use the same underlying (Reachable) abilities [78].

- Test if Observable performance holds under slight perturbations of inputs, changes in prompting style, or minor variations in context. Robustness suggests the observed behavior taps into a stable region of the Reachable space [63, 107].

**Probing the Reachable Model**: The characteristics of $\mathcal{M}_R$, including its potential for beneficial emergent capabilities or undesirable harmful behaviors, should be assessed by investigating the properties of $\mathcal{M}_I$. Rather than passively waiting for behaviors to manifest in the Observable tier, the goal is to proactively use interpretability techniques, or formal methods on both $\mathcal{M}_R$ and $\mathcal{M}_I$ to predict and characterize its potential behavioral repertoire. To this end, consider:

- Develop and apply advanced mechanistic interpretability to probe the $\mathcal{M}_I$'s internals, map its Reachable behaviors and circuits (beyond input-output analysis), acknowledging field limitations and progress [105].

- Employ intervention methods (*e.g.*, from causal inference) to see how internal components affect Reachable behaviors, going beyond simple ablation to controlled perturbations and counterfactual analyses [129].

- Characterize the "behavioral envelope" or "capability manifold" by exploring model responses to diverse, structured inputs aimed at revealing a wide range of latent skills [8, 65].

- Analyze training dynamics and learning trajectories, check if the model learns representations aligned with the instantiated framework, or it finds "shortcuts," exploiting spurious data correlations, and developing misaligned internal concepts [94, 97].

**Assessing the Instantiated Model**: The properties of the $\mathcal{M}_I$ should be evaluated for their fidelity to the specifications, goals, and theoretical underpinnings of $\mathcal{M}_C$. This involves asking: How faithful do interpretations of observed behavior map onto the model's actual internal computations and mechanisms? Are there significant or unintended deviations that arose during implementation or training? The objective is to assess the fidelity of the actual implementation against its original design specifications and theoretical goals. Achieving this requires considering:

- Conduct rigorous audits of the implemented architecture and core algorithmic components (e.g., attention mechanisms, layer structures, activation functions) against the detailed specifications and theoretical assumptions documented in $\mathcal{M}_C$ [80].

- Analyze chosen hyperparameters and embedded architectural constraints for consistency with $\mathcal{M}_C$'s design rationale and implicit theoretical foundations [50, 53, 58, 126].

- Developing techniques to trace the influence of pre-training data [23, 56], initialization strategies, and architectural components [113] on the model's ultimate potential (its Reachable state) is crucial.

- For pre-trained foundation models, assess if their initial representations and zero-shot capabilities on relevant basic tasks align with the objectives and knowledge domains of their conceptual pre-training design [118].

**Validating Alignment with the Conceptual Model**: $\mathcal{M}_C$ itself can be subjected to evaluation by investigating its coherence and soundness, involving philosophical and theoretical critique: How well do the theories of language, reasoning, or intelligence embedded in $\mathcal{M}_C$ align with deeper principles of true linguistic competence or general intelligence? Does $\mathcal{M}_C$ adequately represent the problem it aims to solve or the reality it aims to model? We should therefore consider:

- Engage in critical analysis of the core concepts (*e.g.*, *understanding*, *reasoning*, *creativity*, or *safety*), ensuring their definitions are adequate and well-grounded approximations of the true (Noumenal) nature of these complex phenomena [1].

- Apply principles from the philosophy of information and epistemology, guaranteeing $\mathcal{M}_C$ explicitly acknowledges its own *level of abstraction*, inherent simplifications, and limitations concerning the complexity of the Noumenal ideal it seeks to address [38, 124].

- Subject $\mathcal{M}_C$ to scrutiny from experts in relevant fields beyond AI, such as cognitive science, linguistics, philosophy, and ethics, to assess the validity and potential blind spots of its foundational assumptions [9, 79, 103].

**Acknowledging the Noumenal Model in Evaluation Design**: While $\mathcal{M}_N$ may be unknowable in its entirety, its consideration should inform evaluation design. This means designing evaluations that probe for "core knowledge" or fundamental understanding of underlying principles (*e.g.*, intuitive physics, causality, basic logic), rather than solely testing task-specific pattern matching. The tasks included in $\mathcal{M}_O$ should be critically assessed to determine whether they serve as good proxies for the more fundamental principles believed to constitute $\mathcal{M}_N$ for a given *domain of intelligence*, regardless of how they are conceptualized (*e.g.*, Turing Test [111], Winograd Schema Challenge [62] and their variations [15, 24, 51, 74]).

# 5 Conclusion

The rapid ascent of LLMs has outpaced traditional methods of understanding and evaluating artificial intelligence. We argue that a core issue lies in a persistent mis-cognition of the "model" concept itself, often leading to an over-reliance on narrow, observable behaviors as proxies for overall model quality and capability. To help address this, we organize the historical insights into a proposed five-tiered hierarchical framework, distinguishing between the Noumenal (the ultimate generative principles), Conceptual (the intended design), Instantiated (the algorithmic artifact), Reachable (the full potential behavior space), and Observable (the witnessed behaviors) models, aiming to offer a more cohesive and systematic lens for approaching AI model evaluation. We have explored how LLMs' vast behavioral potential, particularly when viewed through the lens of historically diverging conceptualizations of these tiers in machine learning, challenges established experimental patterns and underscores the value of evolving our evaluation systems. We suggest that developing evaluation strategies that explicitly consider the relationships between these tiers can lead to more insightful and robust assessments. Adopting such hierarchically-informed perspectives is not intended to propose an entirely new paradigm in isolation, but rather to encourage a more nuanced and contextualized approach by building upon the collective and structured understanding of the field. This way of thinking endeavors to cultivate a more meaningful comprehension of these complex systems, fostering responsible innovation and contributing to the development of AI that is beneficial, robust, and aligned with human values.

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

# A    A Feasible Proposal for Explicitly Hierarchical Evaluation Frameworks

Building on our summary of LLM assessment literature in Section 4.2, which highlighted key concerns and practical methods relevant to our suggested layer-wise framework, this section aims to synthesize these elements into a cohesive, operationalizable schema. Hierarchical evaluation frameworks are particularly promising, as exemplified by the ReFeR system [83]. ReFeR employs a "peer review" model where smaller AI "evaluators" provide initial assessments, and a more capable "area chair" model synthesizes these into final scores and reasoning. This hierarchical, multi-perspective meta-methodology strongly aligns with our goals. Consequently, we adopt a similar hierarchical system to embody our proposed *model cartography* as shown below.

- ($\mathcal{M}_O$) The assessment group that focuses on benchmark performance and output quality.

    - One important approach involves employing adaptive testing and dynamic benchmarks, as exemplified by Zhang et al. (2024) [131]. Their method first extracts reasoning graphs from existing benchmark data points and then perturbs these graphs to generate novel test cases. Subsequently, a code-augmented LLM verifies the correctness of labels for the newly generated data. Using this framework, they observed that LLM performance declines with increasing task complexity, often revealing greater biases and excessive sensitivity to specific content. This research highlights the value of evaluating LLMs beyond conventional static benchmarks, offering a more dynamic perspective on assessing their capabilities and limitations.

    - Another promising practice is measuring an LLM's "usage of capacity," as proposed by Cao et al. (2025) [19]. The central concept is that a comprehensive assessment of an LLM's ability should consider the effort it expends to achieve an outcome. To this end, they introduce a measure called MUI to quantify how extensively a model utilizes its capabilities to complete tasks. This approach yields model rankings consistent with expert judgment and demonstrates robustness to variance. Their work offers a significant step towards addressing challenges in assessing model capacity and potentially mitigating the impact of data contamination on evaluations. Furthermore, combining this method with adaptive testing and dynamic benchmarks could allow for a more comprehensive "observation dynamics" of LLM performance.

- ($\mathcal{M}_R$) The assessment group that employs interpretability techniques and evaluates robustness to perturbations.

    - Advanced mechanistic interpretability is vital for understanding and confirming a model's reachable capabilities. For instance, circuit discovery is a key method for linking these observable behaviors to the model's internally instantiated concepts. Seminal work by Elhage et al. (2021) [35] has shown how specific learned modules, like attention heads, are integral to behaviors such as in-context learning. Although applying such detailed techniques to larger, more complex models is challenging, emerging automated methods like Automatic Circuit DisCovery (ACDC) [27] offer a promising path. These tools aim to pinpoint underlying mechanisms and could provide more systematic ways, potentially even quantifiable measures, to assess how specific, predefined behaviors (*i.e.*, a standard dataset) are realized within a model's architecture.

    - Complementing interpretability techniques, exploring model responses to strategically designed inputs provides valuable insights. Adversarial prompting is a prominent example, involving inputs crafted to exploit model vulnerabilities or specific processing mechanisms. The generation of such prompts has become increasingly accessible; for instance, tools like AdvPrompter [89] can rapidly produce human-readable adversarial prompts, sometimes even while preserving the original prompt's semantic meaning. A model's susceptibility and characteristic responses to a suite of such prompts can serve as a crucial basis for metrics to evaluate its propensity for undesirable reachable behaviors, including generating misinformation, revealing sensitive information, or engaging in incorrect reasoning.

- ($\mathcal{M}_I$) The assessment group that investigates the faithfulness of explanations or analyzes internal activation patterns.

- For different instantiations of a concept, it's crucial to assess whether their chosen hyperparameters and architectural constraints align with the underlying design rationale and theoretical foundations. This is particularly true for models with pre-trained initializations, where the pre-training data is fundamental to how well the model instantiates the concept by leveraging its learned general features and representations. These theoretical foundations are often informed by scaling laws [50, 53, 58, 126]. Scaling laws describe how a model's performance (often measured by loss) predictably relates to key factors such as model size, dataset size, and training compute, with this relationship typically characterizable by mathematical functions. Consequently, by comparing the actual performance of a series of model instantiations against predictions derived from these scaling laws, we can confirm the quality of their instantiation. This process also allows us to forecast the potential performance of significantly larger models, thereby guiding strategic research directions.

- Another effective approach to assess how well concepts are instantiated within a model is to employ visualizations of its internal modules. For instance, attention head visualizations [113] illustrate the patterns of attention, detailing how much consideration each token gives to other tokens in the input (or across encoder-decoder interactions) within specific attention heads and layers. This aids in understanding information flow and identifying which tokens influence the representations of others. A further example is the visualization of expert routing in Mixture-of-Experts models [117], which demonstrates how different experts are activated in response to various input tokens. Such examples underscore that visualization is crucial for gaining an intuitive understanding of how a model instantiates concepts and for debugging potential conceptual issues.

- ($\mathcal{M}_C$) The assessment group that oversees the multi-tier assessments, comparing them against the documented goals and assumptions of the concepts.

  - Although directly measuring the inherent "quality" of a concept is challenging, translating qualitative concepts and theoretical underpinnings, such as *understanding*, *reasoning*, *creativity*, or *safety*, into measurable outcomes is essential for genuinely assessing conceptual results. To achieve this, there is a pressing need to develop and implement precise metrics. Such metrics are indispensable for objectively evaluating how effectively observed *phenomena* reflect these carefully defined concepts and whether the system under scrutiny adheres to its acknowledged operational boundaries. This, in turn, enables a more robust and empirically grounded validation of that system's conceptual achievements.

# B  Border Impacts and Limitations

**Border Impacts**: Adopting the proposed five-tier hierarchical view of AI models carries significant implications for AI development, research methodology, and AI epistemology. For development, it encourages designing systems with evaluability in mind at each level, from clear concept articulation and transparent implementation to better management of the optimized model's scope. In research, this framework calls for a shift from "SOTA-chasing" on narrow benchmarks to new experimental designs and metrics that provide deeper insights into different model tiers, their interrelations, behavior, generalization capabilities, and alignment. Epistemologically, the framework redefines what it means to "understand" an AI model, challenging the adequacy of single-score evaluations for complex systems and aligning with broader philosophical discussions about observing and inferring truths about partially unobservable entities, such as those addressed by the Noumenal and Reachable tiers, thereby contributing to the philosophy of AI.

**Limitations**: Operationalizing the proposed multi-tier evaluation framework presents considerable challenges. A primary difficulty is defining clear, measurable, and meaningful criteria for the more abstract Noumenal and Conceptual levels; for instance, assessing the "plausibility" of a Conceptual Model becomes empirically perplexing when tied to an inherently unknowable Noumenon. Another significant hurdle involves characterizing the full, combinatorially explosive spectrum of an LLM's potential behaviors, especially for models with billions of parameters and global representations. Furthermore, existing substantial challenges in LLM evaluation, such as frequent benchmark updates, the high cost of comprehensive assessment, and mitigating data contamination, are likely to be

amplified within such a demanding multi-tier regime. Finally, there's the inherent risk that any new, complex evaluation framework could itself become a target for "SOTA-chasing," diverting efforts to metric optimization without necessarily achieving genuine progress in underlying model quality or understanding.

