# OpenReview forum: "Position: Evaluations Should Acknowledge Model Multifacetedness in the Era of Large Language Models"
_NeurIPS.cc/2025/Position_Paper_Track — Submitted to NeurIPS 2025 Position Paper Track_

### Official Review · Reviewer_TJzX · 2025-07-06

**Significance:** 2
**Presentation:** 3
**Rating:** 3
**Confidence:** 4

**Summary:**

The authors contend that the current manners in which models are evaluated is insufficient, thereby proposing a multi-tiered system to define specific manners in which models should be evaluated based on the end goal or desiderata. The paper goes through a summary of different models and evaluation methods to reinforce their claims.

**Strengths:**

The motivation is clear and the evidence appears to be well supported.

**Weaknesses:**

Some of the questions that the authors raise do appear to be already out-in-the-open questions which researchers have been exploring for quite some time, hence there should be some discussion about methodologies that have already been proposed and why these may fall short of what is necessary to reach the ideals set by the authors.

**Questions:**

Why do the authors specifically limit their framework to the levels proposed (this is not a criticism but rather a way to let the authors reflect and maybe internally reflect on the framework they created)? Is there any gray area for overlap where what is being evaluated could be difficult to judge directly? Is there further granularity that the authors believe is appropriate but may not be fully compatible with the proposed framework?

**Alternative Position:**

Yes, and alternative positions are well-considered and named but not addressed

**Author Identification:**

No.

**Context:**

2

**Discussion:**

3

**Ethics:**

["NO or VERY MINOR ethics concerns only"]

**Position:**

Yes, the paper argues for or against a position related to machine learning.

**Support:**

3

**Thoroughness:**

4

---

### Official Review · Reviewer_Drj5 · 2025-08-09

**Significance:** 3
**Presentation:** 2
**Rating:** 3
**Confidence:** 3

**Summary:**

This position paper addresses the “evaluative crisis” in the era of LLM, arguing that current evaluation method rely on benchmark dataset that create a misleading simulacrum of true model capabilities. To address this, the authors propose a fundamental reconceptualization of the “model” itself through a five-tiered hierarchical framework. This framework deconstructs the model into the Noumenal (MN), Conceptual (MC), Instantiated (MI), Reachable (MR), and Observable (MO) tiers. The paper promotes an ongoing “model cartography” approach, focusing on understanding how each tier relates to.

**Strengths:**

* The proposed "five-tiered hierarchical framework" provides a valuable and structured vocabulary for diagnosing evaluation shortcomings.
* The five-tier structure provides a useful lens for analyzing evaluation gaps, especially the divergence between MR and MO.
* Concept of "model cartography" offers a compelling alternative to the current culture of SOTA-chasing.

**Weaknesses:**

* While the framework is conceptually sound, it offers limited actionable steps for practical implementation.

* A significant conceptual tension exists in the proposal to use dynamic benchmarks and interactive protocols to explore the Reachable space". The authors critique static benchmarks for creating a limited MO. However, any new dynamic protocol, by definition, also creates a new MO. The paper does not sufficiently address how these new, more complex observational windows would be immune to the same fundamental problems of bias, overfitting, or creating a "hyperreal assessment environment" that it critiques.

* Including inherently unknowable tiers (e.g., MN) and aiming to assess all five levels may be unrealistic for the community

**Questions:**

* How can the proposed "dynamic benchmarks" be designed to prevent the formation of a new, equally susceptible MO, thereby avoiding the same "simulacrum" of understanding that the paper critiques in current methods?

**Alternative Position:**

Yes, and alternative positions are trivial straw-man arguments

**Author Identification:**

No.

**Context:**

3

**Discussion:**

2

**Ethics:**

["NO or VERY MINOR ethics concerns only"]

**Position:**

Yes, the paper argues for or against a position related to machine learning.

**Support:**

2

**Thoroughness:**

4

---

### Official Review · Reviewer_2EMv · 2025-08-27

**Significance:** 2
**Presentation:** 2
**Rating:** 4
**Confidence:** 4

**Summary:**

The paper identifies an issue with the evaluation protocol of current LLMs, claiming that current evaluations only assess LLMs through a “small observation window” which may lead to an erroneous understanding of the true model capabilities. Instead, the paper argues for a hierarchical framework of what constitutes a model. This framework is inspired by philosophical ideas (e.g., Kant’s Critique of Pure Reason). The framework includes the Noumenal, Conceptual, Instantiated, Reachable, and Observable models which form a hierarchy that either encompasses or equals the other. The Noumenal model is the true, underlying reality the AI is attempting to approximate, while the observable model is the small window through which current evaluation protocols view the model. The paper argues that we should focus on evaluating and assessing each of these models rather than just the observable model.

**Strengths:**

- Using philosophical ideas and concepts within the machine learning community is an interesting idea that the community can benefit from, especially when working with topics such as LLMs and epistemic questions regarding knowledge, understanding, and truth.

- Broadening the scope of LLM evaluations is an important and significant area that should receive more attention.

- The distinction between the different types of models is interesting.

**Weaknesses:**

- The paper focuses most of its attention on the definitions of the different models and how they differ from each other, and does not provide much detail on how each of these models should be evaluated. While the paper does give general ideas in 4.2 and App. A., they are vague and few.

- The different model types and the differentiation between them are not always clear. I think the writing can be improved to make it more intuitive.

- The paper does not offer an alternative position.

- The paper focuses a lot on the philosophical definitions, and the writing has a philosophical tone to it. For a machine learning conference, I think it has too few machine learning aspects, details, and implications, I believe the paper can be improved by making it more accessible to the wider machine learning community (e.g., while I hold Kant’s Critique of Pure Reason in high regard, I doubt many researchers in the community have read it and I am not sure using Kantian terms helps the paper).

- The paper does not provide any empirical evidence that demonstrates the position is feasible and useful.

**Questions:**

- Is it possible to add a few figures that demonstrate the main ideas of the paper and summarize them?

- Do you have empirical evidence that demonstrates the position is feasible and that using the proposed framework is useful? Alternatively, if no such evidence is present, can the paper clearly explain where and how the framework would be useful?

- Can the authors include a more detailed plan on how to evaluate each of the models and what direct benefit the community would have from doing so?

**Alternative Position:**

No

**Author Identification:**

No.

**Context:**

3

**Discussion:**

2

**Ethics:**

["NO or VERY MINOR ethics concerns only"]

**Position:**

Yes, the paper argues for or against a position related to machine learning.

**Support:**

2

**Thoroughness:**

4

---

### Note · Authors · 2025-09-03

**1-10 Additional Comments:**

We've identified several core challenges with the current Position Paper track:

Flawed Review Standards: Reviewers often judge position papers using technical criteria, prioritizing "usefulness" over the quality of the idea itself. This forces speculative thinking into a practical box and discourages bold, non-traditional ideas.

Lack of Real Debate: Papers are treated more like items on a resume than conversation starters. Reviewers often give safe, non-committal feedback on controversial ideas, stifling genuine academic debate.

Reviewer Mismatch: Position papers on diverse topics like ethics, paradigm critiques, and community building are all lumped together. This leads to reviewers being assigned papers outside their area of expertise.

Barriers for Newcomers: There is a lack of formal channels for discussing and passing on academic skills, making it difficult for students and junior researchers to navigate the field and contribute effectively.

**1-11 Submit Again:**

Probably no

**1-1 Submission Process:**

1

**1-3 Future Development:**

We believe organizers must fundamentally reform the mechanism, shifting from "Gatekeepers" to active "Idea Curators."

First, implement contribution-based submission categories, such as (A) Paradigm Critique, (B) Future Vision, (C) Ethics & Societal Impact, or (D) Community Building & Education. This ensures precise reviews from the start.

Second, establish a "Dedicated Position Paper Review College" through open recruitment, matching experts to submission categories to solve the "mismatched expertise" problem. The review criteria must change. The key question is: "Does this paper propose an idea important, novel, or controversial enough to warrant a dedicated community discussion?"

Third, shift papers from static "credentials" to dynamic conversations. Accepted papers should become "living manuscripts" on a dedicated portal, where authors can update them based on post-conference community feedback, turning the paper from an "end product" into a "starting point."

Fourth, focus on long-term impact. Invite authors of past papers that have sparked significant debate to present at the current conference, rewarding ideas with demonstrable influence rather than just newly accepted ones.

Finally, establish a year-round preprint portal where papers are "post-reviewed" by the community through public discussion and citations. Organizers then act as "curators," inviting the most impactful ideas from this platform to be presented at the conference.

**1-4 Interest:**

["Panel discussions with other position paper authors", "Structured debates on controversial topics", "Workshops for developing position papers", "Mentorship programs for early-career researchers"]

**1-5 Thoughtful:**

3

**1-6 Supportive:**

2

**1-7 Technical Aspects Versus Position:**

2

**1-8 Gate Keeping:**

1

**1-9 Camera Ready Changes:**

1. Change the title and the introduction to make readers less likely to judge its usefulness.
2. Add figures and other easier-to-understand things.
3. Update Section 4 to give a better presentation of how the framework can be employed in the model evaluation case.

**3-1 Review Response1:**

2EMv

**3-2 Reaction To Review1:**

As you rightly pointed out, this article is more of a philosophical discussion, and its purpose is not to provide an immediately useful method to solve the current evaluation problems faced by LLMs. The contribution of this article to the community lies in providing a thinking framework for the various existing papers that claim to solve evaluation problems, allowing researchers to better determine how an evaluation method should be positioned in terms of the "model" concept. With this positioning, the various separately proposed evaluation methods can form a comprehensive and holistic system for understanding models, thereby helping researchers to improve them more effectively. Another purpose of this article is educational and service-oriented: to lay out a clear system for newcomers to the field, thus helping them avoid the influence of the various chaotic definitions currently prevalent in the community.

Furthermore, we do not agree with your view that this article is difficult to discuss. We have described our thought process in detail within the paper, and any questions a reader raises about the framework itself constitute a discussion of the article's viewpoint. For example, in the weaknesses section, you mentioned, "The different model types and the differentiation between them are not always clear." This is a direct discussion of the content of this article, and it is a perfect example of an Alternative Position. We welcome similar discussions, as they are very helpful in maintaining a high level of critical thinking within the community. This is precisely where the article is useful: it can provide better meta-level perspective on research directions. Our future plan is to use this framework as a basis for a discussion on evaluating model evaluations and to extend this framework to other domains.

**3-3 Review Response2:**

Drj5

**3-4 Reaction To Review2:**

We believe this question excellently demonstrates the original intention behind writing this article. Firstly, we did not propose **dynamic benchmarks.** Our intention in mentioning that solution was precisely to describe the narrowness of $M_o$. Secondly, the essence of your question here is defining what a **true $M_o$** is, and this very question represents a genuine **Alternative Position** to our paper. You have mistakenly concluded that our article is proposing a new set of model evaluation methods. This article is intended to provide a discussable framework for the currently chaotic definition of **model evaluation.** When you are using the framework we have defined, this article has already achieved its purpose.

**3-5 Review Response3:**

TJzX

**3-6 Reaction To Review3:**

We believe you may have misunderstood the starting point of our article. Our purpose is not to propose a new set of model evaluation methods. On the contrary, our core contribution is to introduce a framework that can be used when discussing model evaluation. Based on this framework, researchers can align the content of their discussions, rather than each discussing different models in their own context.

The questions you raised, in fact, represent an **Alternative Position** within the scope of this paper. This is because the design of the framework is flexible, and its gray areas can be better defined.

Therefore, we hope you will not simply dismiss this article as discussing an outdated topic. Instead, we encourage you to see it as providing a foundational skeleton for discussion in a field that is currently chaotic.

---

### Meta-Review · Area_Chair_eMPq · 2025-09-19

**Rating:** 3
**Confidence:** 3

**Strengths:**

The paper is commended for its ambitious and intellectually stimulating core idea, which seeks to address a significant perceived shortcoming in how large language models are evaluated. Reviewers found the proposed five-tiered hierarchical framework to be a valuable conceptual tool that provides a structured and nuanced vocabulary for diagnosing the limitations of current benchmark-based evaluation methods. The philosophical grounding of the work, while also a point of concern, was simultaneously seen as an interesting and potentially beneficial approach for the machine learning community, particularly when grappling with epistemic questions of knowledge and model capabilities. The overarching goal of moving beyond a narrow "observable model" towards a more comprehensive process of "model cartography" was viewed as a compelling and worthwhile alternative to the prevailing culture of chasing state-of-the-art leaderboard results.

**Weaknesses:**

The primary weakness of the paper lies in the significant gap between its high-level conceptual proposal and its practical implementation. The framework is criticized for being vague and offering limited actionable steps for how the community could actually evaluate the different tiers, especially the inherently unknowable "Noumenal" model. A major conceptual flaw was identified in the proposed solution of using dynamic benchmarks, as any new protocol would itself create a new "Observable Model" window, potentially replicating the very same problems of bias and simulacrum that the paper critiques in current methods. Furthermore, the writing is seen as overly philosophical and inaccessible for a broad machine learning audience, with too much focus on definitions and too little on empirical evidence, concrete machine learning details, or a discussion of how this approach improves upon existing research that already explores similar evaluation challenges.

**Questions:**

1. how can their proposed dynamic benchmarks be designed to fundamentally avoid the pitfalls of current static ones, preventing the creation of a new, equally problematic observational window?

2. what concrete, empirical evidence can be provided to demonstrate the feasibility and utility of this framework, moving beyond pure philosophical argumentation?

3. the authors are asked to justify the specific boundaries and granularity of their chosen tiers, reflect on potential areas of overlap between them, and explain how their novel framework specifically advances the field beyond methodologies that researchers are already exploring.

**Thoroughness:**

3

---

### Decision · Program_Chairs · 2025-09-26

Reject